# Self-reported menses physiology is positively modulated by a well-formulated, energy-controlled ketogenic diet vs. low fat diet in women of reproductive age with overweight/obesity

**Madison L. Kackley**[1☯], **Alex Buga**[1☯], **Milene L. Brownlow**[2], **Annalouise O'Connor**[2], **Teryn N. Sapper**[1], **Christopher D. Crabtree**[1], **Bradley T. Robinson**[1], **Justen T. Stoner**[1], **Drew D. Decker**[1], **Loriana Soma**[3], **Jeff S. Volek**[1]*

1 Department of Kinesiology, The Ohio State University, Columbus, Ohio, United States of America,
2 Metagenics, Inc., Aliso Viejo, California, United States of America, 3 Department of Obstetrics and Gynecology, The Ohio State University, Columbus, Ohio, United States of America

☯ These authors contributed equally to this work.
* volek.1@osu.edu

## Abstract

Weight loss can positively alter female physiology; however, whether dietary carbohydrate- or fat- restriction confer unique effects is less studied. Precisely designed, hypocaloric well-formulated ketogenic diets (KD; ~75% energy for weight maintenance) were compared to isocaloric/isonitrogenous low-fat diet (LFD) on self-reported menses in pre-menopausal overweight and obese women (mean ± SD: 34 ± 10 years, BMI: 32.3 ± 2.7 kg/m2). Women received a precisely-weighed and formulated KD with either twice-daily with ketone salts (KS; n = 6) or a flavor-matched placebo (PL; n = 7) daily for six-weeks. An age and BMI-matched cohort (n = 6) was later assigned to the LFD and underwent the same testing procedures as the KD. Self-reported menses fluctuations were assessed bi-weekly along with measures of body weight, body composition, and fasting serum clinical chemistries using repeated measures ANOVA with Bonferroni post-hoc corrections. Both diets elicited clinically-significant weight-loss (Δ: -7.0 ± 0.5 kg; p < 0.001), primarily from fat-mass (Δ: -4.6 ± 0.3 kg; p < 0.001), and improved insulin-sensitivity and serum lipids (all p < 0.05). Fasting plasma glucose and inflammatory markers were not different between diets. Fasting capillary beta-hydroxybutyrate (R-βHB) increased significantly during the KD, independent of supplementation (Δ: 1.2 ± 0.3 mM R-βHB; p < 0.001). Women randomized to the KD+KS (30%) and KD+PL (43%) reported subjective increases in menses frequency and intensity after 14 days, whereas another third reported a regain of menses (>1 year since the last period) after 28 days. No LFD participants reported menses changes. Nutrient-dense, whole-food KDs and LFD improved weight, BMI, body composition, and blood parameters in pre-menopausal women after six-weeks. Changes in self-reported menses were described by most of the KD participants, but none of the LFD women suggesting there may be unique effects of nutritional ketosis, independent of weight loss.

**Data Availability Statement:** All relevant data are within the paper and its Supporting Information files.

**Funding:** This work was funded by a research project grant received from Metagenics, Inc. The study sponsor was responsible for providing the study products (KS, MCT and protein powder samples), and contributed to study design, but was not involved in data collection, sample analysis, or result reporting.

**Competing interests:** JV receives royalties for low-carbohydrate nutrition books; is founder, consultant, and stockholder of Virta Health, Inc. and is a member of the advisory boards for Simply Good Foods. The remaining authors declare that the research was conducted in the absence of any commercial or financial relationships that could be construed as a potential conflict of interest. This does not alter our adherence to PLOS ONE policies on sharing data and materials.

## Introduction

Excess adiposity, primarily stored as visceral adipose tissue, can alter healthy physiology, specifically glucose metabolism. The arising metabolic sequelae manifest as fasting hyperglycemia (> 100 mg/dL), hypertriglyceridemia (> 150 mg/dL), and impaired insulin signaling–features that describe the early onset of insulin-resistance [1]. Approximately 4-in-10 adult women in the US have been diagnosed with pre-diabetes or diabetes [2], an insulin-resistant condition that increases their propensity for developing sex-specific endocrinopathies (i.e., polycystic ovary syndrome) [2], irregular menses, amenorrhea, and infertility [3]. Therefore, prophylactic lifestyle changes such as diet modifications merit further examination [4, 5].

A well-formulated ketogenic diet (KD) is a nutrient-dense, whole-food lifestyle approach primarily characterized by its unique ability to increase blood ketones into a safe, and potentially therapeutic, range of nutritional ketosis (0.5–4.0 mM beta-hydroxybutyrate; $R$-βHB) [6]. Ketosis is rapidly inducible (~3–5 days) [7] when carbohydrates are limited and primarily derived from non-starchy vegetables (20–50 g/day), protein is consumed in moderation (1.2–1.6 g/kg BW/day), and lipids fulfill the daily energy requirements and satiety. If sustained over time, ketosis can modulate blood parameters positively, both during weight-loss [5, 7–16] and weight-maintenance [16, 17]. Moreover, augmenting a KD with exogenous ketone supplements can plausibly exert positive and dose-dependent effects on glycemia [7, 18–20], thus likely to provide additional health benefits.

Considerable efforts have been directed towards examining low-fat diets (LFD) in the context of women's health [21], specifically in post-menopausal women with insulin-resistance [22]. Although long-term LFD adherence can improve global metabolic parameters in women who are undergoing concurrent weight-loss, conflicting evidence suggests that these recommendations are not a panacea [23]. More recently, KDs have been suggested as a promising therapeutic intervention for reversing metabolic sequalae and endocrinopathies, specifically in women [24], with pilot studies supporting the evidence for improvements in glycemic control, insulin sensitivity, cholesterol profiles, hormonal profiles and circulating androgens [12, 25]. Few studies have carefully examined KD applications in women [23], and even fewer used precise-feeding methods [25, 26]. In general, alternative dietary strategies are increasingly investigated [6, 23, 27, 28], whereas in some cases, they have been prescribed with caution in women [26], implying that more precise work is required to identify the roles of KDs in female physiology, specifically related to pre-menopausal outcomes.

The female-cohort presented herein was extracted from a larger investigation examining the effects of precisely-fed, hypoenergetic KDs, with or without exogenous supplements, versus LFDs in both men and women with overweight and obesity [29–33]. All women from the larger study were included in this analysis. There were three diet conditions: 1) KD augmented with twice-daily ketone salts; 2) KD with a placebo; and 3) USDA-guideline LFD that was isoenergetic-isonitrogenous to KDs. The study lasted six-weeks and precisely-cooked all the meals in a metabolic kitchen. The exploratory results included changes in body weight, body composition, ketonemia, glycemia, insulin and insulin resistance (HOMA-IR), blood lipids, inflammatory markers, and self-reported changes in menses.

## Methods

### Ethical approvals

This study was approved by the Institutional Review Board of Ohio State University–Department of Human Sciences (IRB # 2017H0395). Protocol can be found in S1 File, in accordance with the latest version of the Declaration of Helsinki (2013), except for registration in a

database. Prior to enrollment, all women were fully informed of any risks and discomforts associated with the experiments before consent.

## Participants

Detailed experimental protocols regarding participant randomization, recruitment, enrollment, and other methodology have been previously described [29–33] and a flowchart can be found in Fig 1 and consort checklist S2 File. In brief, pre-menopausal women (mean (SD); Age: 34 (9); BMI: 31.6 (1.5)) were recruited from the greater Columbus, Ohio area to join a weight-loss study exploring the effects of hypoenergetic KD feeding (75% of energy to maintain weight), supplemented with and without daily exogenous ketones. A sample of the KD menu is shown below (Fig 2). Both KDs provided approximately 40 g/day of carbohydrates, and the rest of the non-protein calories came from fats, with an emphasis on monounsaturated and saturated fat sources. The LFD contained 25% of energy from lipids, with less than 10% being saturated fat and less than 30 g of added oils. Carbohydrates in the LFD were mainly complex and included at least 32 g of fiber per day, with limited added sugars (less than 25 g). Meal plans were developed to include a wide range of high-quality protein sources to be distributed equally between breakfast, lunch, and dinner. Detailed nutrient composition can be found in ST1 n S3 File. Nineteen women were enrolled and completed the experiment between

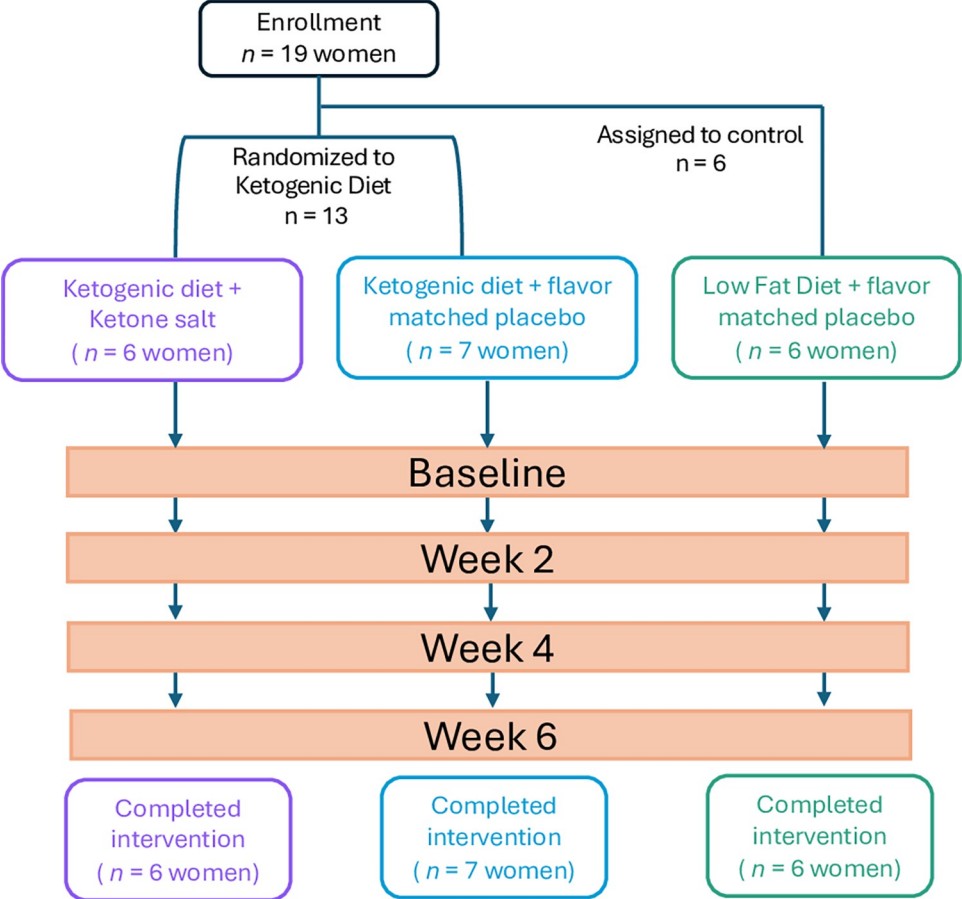

**Fig 1. Participant flowchart.** 19 women were enrolled to the current trial. 13 women were randomized into the ketogenic interventions and later 6 women were assigned to the control.

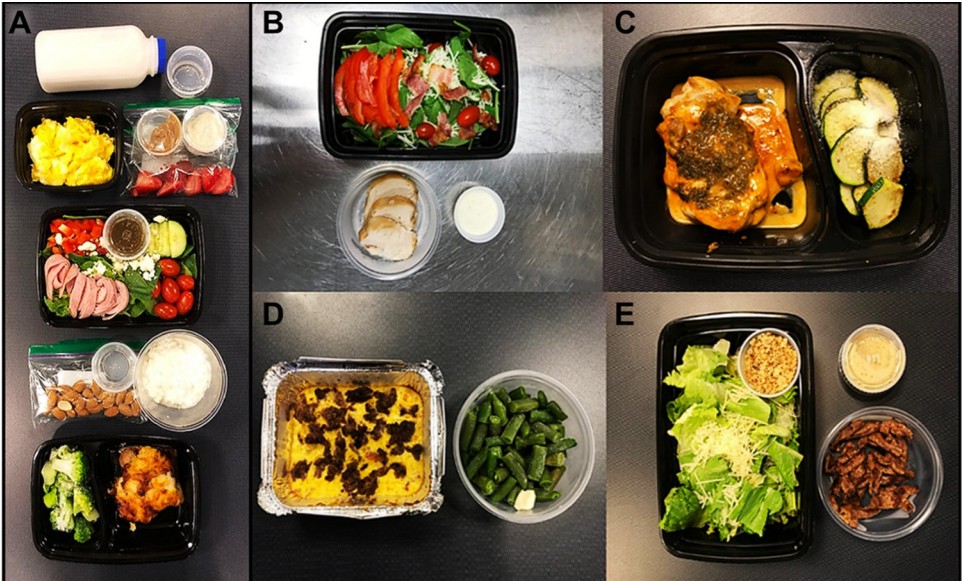

**Fig 2. KD menu.** Food was precisely weighed (± 0.1g) and tailored to the participants' individual energy requirements to lose weight (~75% of the energy maintenance). Panels: A) The breakfast, lunch, snack, and dinner served to participants on bi-weekly test days; B) Chicken Breast Tomato Bacon Salad with Hand-made Ranch Dressing; C) Seared Caribbean-Jerk Chicken Thigh with Parmesan Zucchini; D) Beef, Egg, and Cheese Casserole with Green Beans; E) Flank Steak Salad with Hand-made Low-Carb Parmesan Croutons and Ranch Dressing.

June 2018 and August 2019 at study end. Thirteen women were randomized and balanced to receive either a ketogenic diet + ketone salt supplement (KD + KS, $n = 6$) or ketogenic diet + flavor-matched placebo (KD + PL, $n = 7$). After completion of the KD conditions, we enrolled a third, non-randomized cohort of age and BMI-matched women ("control group") who received an isoenergetic/isonitrogenous low-fat diet (LFD, $n = 6$) and participated in the same testing as the KD groups. Two participants–one in the KD+KS diet and one in the KD +PL–were concurrently taking oral contraceptives before and during the study. The rest of the cohort had no contraceptives to report at day 1.

Exclusion criteria comprised: major weight loss events (< 10% body mass) six months prior to enrollment; habitually consuming a low-carbohydrate diet (< 50 g CHO/day); pre-existing gastrointestinal disorders or food allergies; excess alcohol consumption (> 14 drinks/week); disease conditions (diabetes, liver, kidney, or other metabolic or endocrine dysfunction); use of diabetic medications. Women who met the qualifying criteria were scheduled for an in-person screening meeting where they completed additional questionnaires about medical history, physical activity, and a menstrual history survey. There were no significant differences in baseline characteristics between groups (Table 1).

### Experimental design

This intervention was six-weeks long and involved attending bi-weekly in-person visits. Participants arrived at the testing center (PAES Building, The Ohio State University, Columbus OH) between 6:00–8:00 am, while fasted (> 8h since the last meal), rested (> 7h sleep/night) and hydrated (< 1.025 urine specific gravity). Weight and height were measured upon arrival while wearing light clothing and no shoes. Body composition (regional fat mass, fat-free mass, body fat percentage) was assessed using dual x-ray absorptiometry (GE Lunar Medical Systems, Madison, Wisconsin, US).

**Table 1. Baseline characteristics (n = 19).**

| | Ketogenic Diet + Ketone Salts (KD+KS, n = 6) | | Ketogenic Diet + Placebo (KD+PL; n = 7) | | Low-fat Diet (LFD; n = 6) | | One-way ANOVA |
|---|---|---|---|---|---|---|---|
| | **Mean** | **SD** | **Mean** | **SD** | **Mean** | **SD** | **p-value** |
| Body Weight (kg) | 80.4 | (3.4) | 90.2 | (3.2) | 87.2 | (3.4) | 0.139 |
| Body Mass Index (BMI; kg/m²) | 31.3 | (1.0) | 33.2 | (0.9) | 30.2 | (1.0) | 0.112 |
| Height (cm) | 106.5 | (3.1) | 166.4 | (5.1) | 168.5 | (6.1) | 0.5346 |
| Waist (cm) | 90.7 | (2.0) | 92.0 | (1.9) | 87.3 | (2.0) | 0.259 |
| Hip (cm) | 108.8 | (2.8) | 117.6 | (2.6) | 112.2 | (2.8) | 0.090 |
| DXA Fat Mass (kg) [‡] | 32.7 | (2.5) | 38.9 | (2.3) | 33.6 | (2.5) | 0.168 |
| DXA Fat Free Mass (kg) [‡] | 44.8 | (1.7) | 47.2 | (1.6) | 50.7 | (1.7) | 0.073 |
| DXA Body Fat (%) [‡] | 40.3% | (1.8%) | 43.4% | (1.7%) | 37.6% | (1.8%) | 0.098 |

[‡] = measured using dual x-ray absorptiometry

## Blood chemistry

Blood draws were performed by a trained phlebotomist using aseptic techniques involving antecubital puncture via butterfly needle (21G BD Vacutainer®, NJ, USA). Venous blood was collected in plasma-EDTA and serum-separator vacuum tubes (BD Vacutainer®, 10mL, NJ, USA). Serum was shipped the same-day to an off-site laboratory for lipid analyses, which comprised of total cholesterol (TC), calculated high- and low-density lipoproteins (HDL-/LDL-C), triglycerides (TG), TC-to-HDL-C ratio (TC:HDL-c), and non-HDL-C (Standard Lipid Panel #7600, Quest Diagnostics, OH, USA). Plasma insulin, interleukins (IL-1β, IL-6, IL-8, IL-10), monocyte chemoattractant protein 1 (MCP-1), tumor necrosis factor a (TNF-a) concentrations were determined using enzyme-linked immunoassay (U-PLEX, Meso Scale Diagnostics, MD, USA). Intra- and inter-coefficients of variation were within normal parameters ($< 10\%$). Daily blood ketones and glucose were measured enzymatically in capillary blood using a portable glucometer fitted for glucose and ketone ($R$-βHB) strips.

## Menses survey

The lifestyle changes survey was an exploratory instrument designed to capture subjective perceptions of menstrual events, such as frequency and intensity changes after starting the experimental diets. Day 1 was normalized to "no change" (i.e., baseline). On days 14, 28, and 42, participants were asked if they had noticed any changes in their menses since the last test visit. If "Yes" was selected, the answer implied that the participant observed changes that were noticeably greater relative to their menses month phase; "No" reflected no new changes from the prior visit. When participants answered "Yes", subsequent options were displayed. These secondary options asked if the change can be described as differing in "intensity," "frequency," or if menses was altogether "regained after $\geq 1$ year" (i.e., secondary amenorrhea).

## Statistics

The analyses were conducted using a commercially-available statistics package (IBM SPSS Statistics for Windows, Version 28.0. Armonk, NY, USA). Data was screened for homogeneity of variance and normality using Levene's test and Shapiro-Wilk test, respectively. A one-way analysis of variance (ANOVA) revealed no between-diet differences at baseline except for plasma insulin that was higher in LFD relative to KD. The self-reported answers from the menses survey were assigned into 4 categories: "no change";"change in frequency"; "change in

intensity"; "regained period after >1 year." The four possible answers were graded as binary categories (yes/no) and assigned values (yes = 1; no = 0). A 3 (condition) by 4 (time) repeated measures (RM) ANOVA with a Bonferroni post-hoc correction was applied to all variables, including survey responses; insulin was analyzed using a repeated measures analysis of covariance (ANCOVA) to account for baseline differences (i.e., change scores were analyzed instead of absolute values).

## Results

### Ketosis

Capillary $R$-βHB concentrations were similar between diets at day 1. An interaction effect (all $p < 0.05$) revealed that KDs elevated $R$-βHB beyond the LFD ($p < 0.001$), and into the range of nutritional ketosis (> 0.5 mM $R$-βHB) throughout the experiment. The mean $R$-βHB change between the KD+KS and KD+PL was non-significant (1.2 ± 1.2 vs 1.2 ± 0.7 mM), and neither was peak $R$-βHB (Δ: 1.5 ± 1.2 vs. 1.6 ± 1.0 mM). The LFD did not influence $R$-βHB concentrations (Fig 3).

### Menses

The menses questionnaire was designed to capture changes between bi-weekly experimental visits; therefore, all the responses at day 1 were normalized to "no change."

A 3 (diet) by 4 (time) RM ANOVA revealed significant main effects of time in all menses response categories, however, the "no change" (p = 0.002) and "change in intensity" (p = 0.026) answers were the only categories different compared to Day 1. For the main effect of diet, the "no change" answers in LFD condition were significantly different compared to KD+KS (p = 0.002) and KD+PL (p = 0.008). In other words, there had been at least one significant change in menses that occurred during KD+KS and KD+PL. Despite a significant main effect of diet for "change in intensity," there were no significant post-hoc effects to report. The

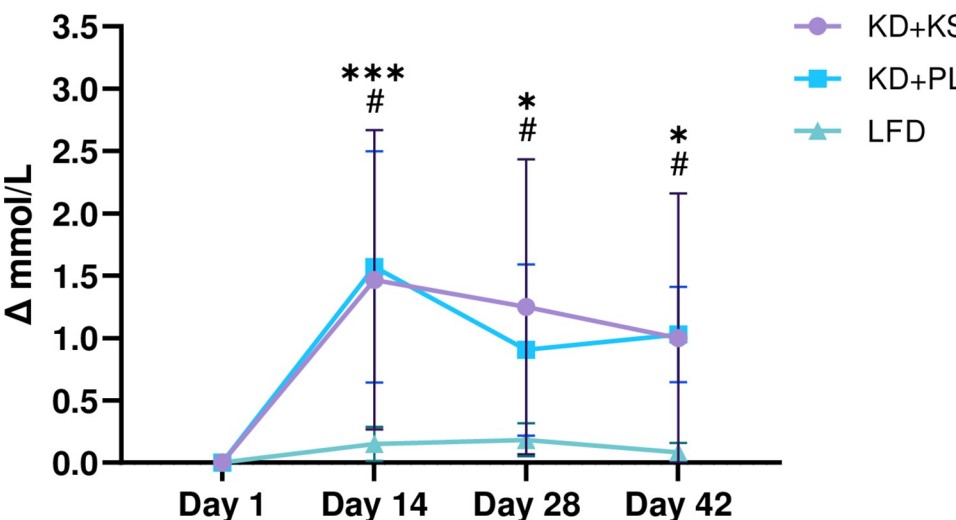

**Fig 3. Change in blood ketones.** Capillary $R$-βHB was analyzed bi-weekly and while fasting. Data presented as mean ± SD. Main effect: *, *** = $p < 0.05$, 0.001 from Day 1 in KD only. Interaction: # = $p < 0.05$ between KD+KS and KD+PL versus LFD at the indicated timepoint.

**Table 2. Menses frequency questionnaire.**

| Survey Answer | Visit | Number of Answers | | | % of Test Visit Answers | | | 3 x 4 RM ANOVA | | |
|---|---|---|---|---|---|---|---|---|---|---|
| | | KD+KS | KD+PL | LFD | KD+KS | KD+PL | LFD | Time | Diet | Interaction |
| No Change | Day 1 | 6 | 7 | 6 | 100% | 100% | 100% | **0.001** | **0.001** | **0.009** |
| | Day 14 | 5 | 5 | 6 | 71% | 63% | 100% | | | |
| | Day 28 | 1[#] | 2[#] | 6 | 11% | 20% | 100% | | | |
| | Day 42 | 1[#] | 4 | 6 | 13% | 57% | 100% | | | |
| Change in Frequency | Day 1 | 0 | 0 | 0 | 0% | 0% | 0% | **0.014** | **0.097** | **0.222** |
| | Day 14 | 1 | 1 | 0 | 14% | 13% | 0% | | | |
| | Day 28 | 3 | 3 | 0 | 33% | 30% | 0% | | | |
| | Day 42 | 3 | 1 | 0 | 38% | 14% | 0% | | | |
| Change in Intensity | Day 1 | 0 | 0 | 0 | 0% | 0% | 0% | **0.042** | **0.050** | **0.257** |
| | Day 14 | 1 | 2 | 0 | 14% | 25% | 0% | | | |
| | Day 28 | 2 | 2 | 0 | 22% | 20% | 0% | | | |
| | Day 42 | 4 | 2 | 0 | 50% | 29% | 0% | | | |
| Regained Period after >1 year | Day 1 | 0 | 0 | 0 | 0% | 0% | 0% | **0.001** | **0.141** | **0.057** |
| | Day 14 | 0 | 0 | 0 | 0% | 0% | 0% | | | |
| | Day 28 | 3 | 3 | 0 | 33% | 30% | 0% | | | |
| | Day 42 | 0 | 0 | 0 | 0% | 0% | 0% | | | |

KD, ketogenic diet; KS, ketone salts; PL, placebo; LFD, low-fat diet

Interaction

= p < 0.05 comapared to LFD

significant interaction effect (diet * time) for "no change" further revealed that responses relative to LFD were significantly different for both KD+KS (p = 0.004) and KD+PL (p = 0.011) at Day 28, and just KD+KS at Day 42 (p = 0.007) (Table 2). There were no other significant interactions to report. A graphical representation of the survey responses is presented below (Fig 4).

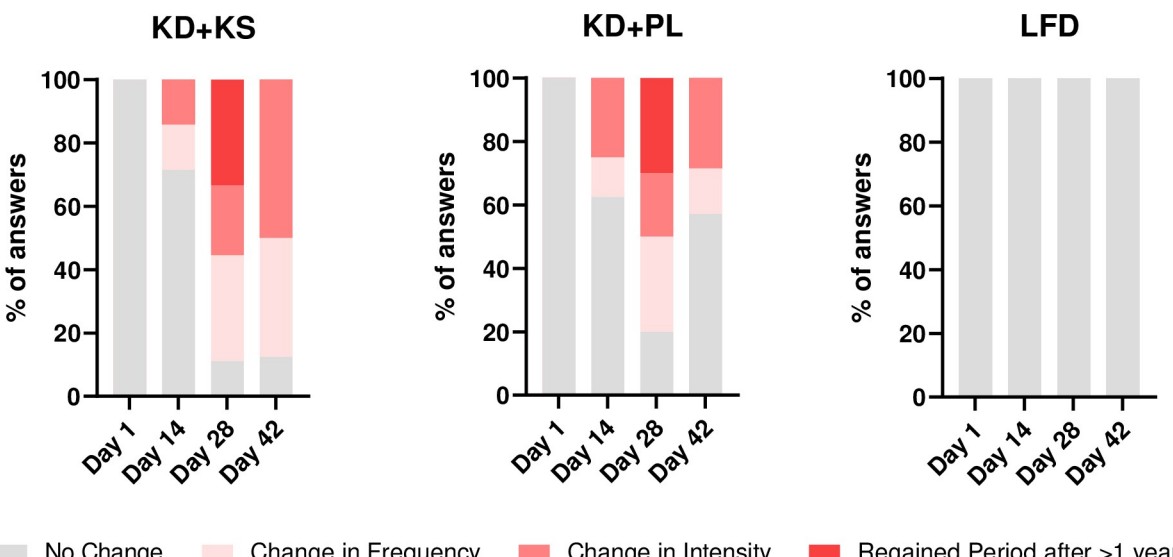

**Fig 4. Cumulative percentage of self-reported menses change answers.** The interaction effect for "no change" revealed that the KD+KS responses at day 28 (*p* = 0.004) and 42 (*p* = 0.007), and the KD+PL responses at day 28 (*p* = 0.011), were significantly different compared to the frequency of answers in the LFD at the same corresponding timepoints.

To assess potential distinctions between women who experienced a modification in menstrual patterns ("responders") and those who did not ("non-responders"), we conducted a Mann-Whitney analysis on baseline characteristics (see S1 Table in S3 File). Responders exhibited a notably elevated BMI compared to non-responders (33 ± 3 vs. 30 ± 2 kg/m2; p = 0.04) and marginally higher serum MCP-1 levels in responders (p = 0.064). However, no other baseline characteristics were found to be indicative of menstrual responses.

To determine if there were any differences between women who reported a change in menses ("Reported Change") versus those who did not ("No Reported Change"), we performed an independent t-test on baseline characteristics (ST 2). Women who reported change had a significantly higher BMI than those who did not report change (33 ± 3 vs. 30 ± 2 kg/m$^2$; $p$ = 0.04). There was a trend for higher serum MCP-1 in women who reported change ($p$ = 0.064), but no other baseline characteristics were different between those who reported change and those who did not.

## Physiological outcomes

Descriptive tables and figures are presented in ST3, ST4 in S3 File, SF1 and SF2 in S4 File. There were significant main effects of time for total body weight loss (Δ: -7.1 ± 2.1 kg; $p < 0.001$) and BMI (Δ: -2.6 ± 0.7 kg/m$^2$; $p < 0.001$) and no significant main effects of diet or diet x time interactions. In-person visit assessments revealed a mean bi-weekly body weight loss rate of ~2.4 kg (2.6% of body weight). The women who were randomized to the KD+PL diet attained clinically meaningful weight-loss (weight loss of ≥ 5% from baseline) at day 14, and by day 28 on the KD+KS and the LFD achieved clinically meaningful weight loss. Out of the fourteen women who started with an obesity class I BMI at baseline (> 30 kg/m$^2$), nine (~64%) reversed to an overweight BMI (< 30 kg/m$^2$) by the end of the intervention.

Weight loss primarily originated from fat mass losses (Δ: -4.6 ± 1.3 kg; $p < 0.001$), with a minor but significant portion from fat-free mass (Δ: -1.7 ± 1.7 kg; $p$ = 0.002). Overall body composition improved as reflected by total body fat percentage (Δ: -3.6 ± 4.9%; $p$ = 0.033). The bi-weekly rate of fat mass losses was 1.5 kg and was significant between each experimental visit ($p < 0.001$). Fat-free mass losses were primarily driven by the changes from day 1 to day 14 (Δ: -1.3 ± 0.7 kg; $p < 0.001$), followed by a non-significant plateau thereafter. For every kilogram of body weight lost, 73% was derived from fat mass and 27% from fat-free mass (~2.7:1 fat:fat-free mass).

## Blood parameters

Plasma glucose was not affected by time or diet. Plasma insulin was significantly higher at day 1 in the LFD relative to the KD+KS (17.6 ± 7.0 vs. 8.4 ± 1.9 µU/mL; $p$ = 0.02), therefore, normalized values (i.e., change from baseline) were analyzed for significant effects. The main effect of time from fasting insulin at day 1 to day 42 (Δ: -4.2 ± 3.6 µU/mL; $p < 0.001$) manifested as a sharp reduction at day 14 and sustained thereafter. The significant diet effect revealed that mean fasting insulin was lower during the LFD versus the KD+KS (LFD–KD+KS: -5.6 ± 6.6 µU/mL; $p$ = 0.006). The HOMA-IR index was not affected by diet; however, weight loss improved insulin-sensitivity significantly from day 1 to day 42 (Δ: -0.9 ± 0.9; $p < 0.001$).

There were no significant diet differences for TC, TG, LDL-C, HDL-C, TC:HDL-C, or non-HDL-C concentrations. Wherever time effects were significant, all participants were grouped and analyzed together ($n$ = 19). Time differences from day 1 to day 42 were significant for TC (Δ: -28.3 ± 32.9 mg/dL; $p$ = 0.01) and HDL-C (Δ: -10.4 ± 13.9 mg/dL; $p$ = 0.03). The LDL-C concentrations decreased significantly from day 1 to day 28 (Δ: -11.3 ± 15.8 mg/dL; $p$ = 0.039),

but not compared to day 42 (Δ: -9.5 ± 20.6 mg/dL; $p$ = 0.37). Similarly, non-HDL-C concentrations decreased at day 28 (Δ: -15.4 ± 17.0 mg/dL; $p$ = 0.007), whereas the effect was no longer significant by day 42 (Δ: -13.2 ± 22.2 mg/dL; $p$ = 0.12). The significant time effect for TG and significant interactions for LDL-C and non-HDL-C returned no significant post-hoc comparisons.

Weight loss did not influence inflammatory markers. A main effect of diet was obtained for IL-1β, with greater concentrations detected during the KD+KS intervention versus the KD +PL (Δ: 0.10 ± 0.07 pg/mL; $p$ = 0.011) and the LFD (Δ: 0.12 ± 0.06 pg/mL; $p$ = 0.005), an effect largely attributed to baseline concentrations. The significant interaction effect for TNF-α returned no significant post-hoc comparisons (ST 4).

## Discussion

### Brief summary

Nineteen women successfully completed six-weeks of precise hypocaloric feeding. The KD, with or without exogenous ketones, elicited clinically significant weight-loss, improved body composition, reduced cardiometabolic risk, and augmented insulin sensitivity to the same extent as the LFD, suggesting that there are no significant differences in these short-term health outcomes between diets varying widely in macronutrient distribution when prescribed at a caloric level designed to induce weight loss. There were, however, significant differences between the KD and LFD in self-reported menses with most women assigned to the KD reported at least some change in menses over the six-week intervention, whereas no participants in the LFD group reported a change in menses. This novel finding suggests that independent of weight-loss, there may plausibly be something unique to the KD, such as elevated $R$-βHB, that influences menses physiology.

### Self-reported changes in menses

The fact that menses modulations during the intervention were exclusively reported within the KD condition, but not LFD, reflected a novel and specific outcome of carbohydrate-restriction. Previous literature generalizes body weight as an independent modulator of menstrual cycle regularity [34], an effect confirmed by dietary interventions seeking to attain clinically significant weight-loss women, independent of diet composition [35]. In our intervention, 11 out of 13 women enrolled to the KD (~85%) reported at least one change in either menses frequency, intensity, or both, at some point during the intervention, whereas no change was reported throughout the LFD. Six women in the KD condition–three in each KS and PL–reported a regain of menses after not having a cycle for > 12 months (i.e., operational definition for secondary amenorrhea); three of them were >35 years old. Moreover, it is important to mention that the two women in the KD who did not report a change in menses were the ones who reported concurrent use of birth control (both reported using oral contraceptives), thereby all the effects that we detected occurred exclusively in non-birth-control users. The findings highlight that adoption of a KD by women is associated with self-reported menses changes over six-weeks, which cannot be explained by weight loss since a matched group of women consuming an isocaloric LFD reported no changes in menses, despite comparable weight loss and metabolic changes.

### Weight-dependent effects

We demonstrated that reversal of overweight and obese phenotypes in an otherwise-healthy and pre-menopausal cohort can be attained using two different diet modalities, remarkably as

early as two-weeks. The overall changes from day 1 to day 42, such as total percent body weight change (-8%), BMI (-8%), body fat percentage (-9%), fasting insulin (-34%), HOMA-IR (-34%), TC (-15%), confirmed our original hypothesis regarding weight as a major determinant of holistic improvements in women physiology.

Our precisely-feeding approach was sensible to inclusion of whole-foods and nutrient-density, a detail rarely employed in assessing the effects of diet on women physiology. A 4-week cross-over trial, that was similar to our clinically-controlled feeding strategy–albeit weight-stable–compared the effects of a KD to a standard diet (NFA Swedish Guidelines) in 17 healthy young women (median age: 23.8 years) to assess the effect of diet on blood parameters, specifically lipids [26]. Their results revealed a significant effect within KD that resulted in increasing TC after 4-weeks (Δ: 41 mg/dL), directly explained by the changes in LDL-C (Δ: 33 mg/dL) and HDL-C (Δ: 6 mg/dL), plus an additional increase in TG (Δ: 2.3 mg/dL) and decreases in fasting glucose (Δ: -9 mg/dL) and insulin. Compared to our results, we did not observe significant changes within KD participants that would suggest deleterious effects of diet, largely because our results were primarily influenced by the significant weight-loss, therefore, a diet-specific effect cannot be ascertained. However, our results revealed no change or a decrease in cardiovascular risk parameters from day 1 and thereafter, effects that were predominantly influenced by weight changes and the magnitude of change induced by the LFD. Collectively, we observed that both the KD and LFD can be used to attain clinically significant weight loss without raising undue cardiometabolic risks in women; however, future work is encouraged to use precise, non-ad-libitum feeding designs to accurately assign changes in blood lipids to diet.

## Weight-independent effects

The fact that we were able to detect menses events in KD but not the LFD was an unexpected and unique outcome conferred by diet. Whereas subjective changes in menses intensity could have been obfuscated by inaccurate self-assessment, the changes in menses frequency and regain of menses were less likely to be confounded by personal observations due to increased objectivity (i.e., subjective appearance vs. objective occurrence). In other words, we likely captured greater-than-random changes in self-reported menses with our brief survey. Moreover, by comparing the women in the KD+KS and KD+PL to LFD, we were able to precisely identify when and how this weight-independent effect occurred. It is unclear how the changes can be ascribed to KD or $R$-βHB, thus more work is required to probe deeper into the mechanisms that explain these effects.

## Future work

The weight-loss literature has pointed at implementing the KD for women diagnosed with PCOS. The PCOS evidence overwhelmingly supports weight-loss as an effective strategy to improve health, whereas KD sustained over longer durations provided additional benefits. A previous 12-week pilot trial has revealed that a KD with no pharmaceutical assistance can regulate menses positively, and to the same extent as a mixed diet with pharmaceutically-treated PCOS, whereas the glucose control and body weight changes favored the KD [9]. A single-arm, 12-week trial conducted in 14 women with PCOS also revealed that the KD is a feasible, non-pharmacological intervention to improve body weight, body composition, insulin-sensitivity, cardiometabolic risk, and female-specific circulating hormones [12]. Lastly, a 24-week single-arm study examining the KD effects on metabolic and endocrine parameters enrolled 11 women with PCOS– 5 completing all study details (~45% retention)–who demonstrated positive reductions in testosterone, LH/FSH, and fasting insulin by the end of the diet [10].

The next step is to design larger studies ($n > 30$) that capture diet-induced effects on hormones and metabolites associated with women's health. The goal is to use our present results as a priori justification for larger sample size and blood analyses in future investigations. Secondly, we are motivated by our findings to continue researching the effects of KD in women, specifically improving insulin-resistance, and the pleotropic effects on symptoms of PCOS and perimenopause. The main goal is to elucidate how (and if) diet or $R$-βHB modulate ovulation and reproductive health by measuring global physiological and metabolic parameters that can be included in multivariate regression analyses. Given the plausible interaction between $R$-BHB and self-reported change in menses that we observed in this investigation, the mechanisms merit further attention as they relate to women's physiology, specifically fertility.

### Strengths/Limitations

A major strength of our intervention was the precise feeding. Compared to dietary records and recalls (i.e., 24h, 3-day recall), preparing all the food in a metabolic kitchen removed the ambiguity of self-reporting and enabled us to make precise assumptions about the observed changes. Moreover, due to the inherent features of the KD, the adherence to the diet was bolstered by capillary finger-stick tracking of $R$-βHB ($> 0.5$ mM), whereas LFD adherence was assumed from weight trends and the food containers returned to the laboratory (i.e., 100% of the food containers were empty for both the LFD and KDs when returned to the testing facility).

One potential limitation was not including a female-specific blood panel in our original analysis plan, such as estrogen, progesterone, sex-hormone binding globulin, luteinizing hormone, or follicle-stimulating hormone. A secondary limitation would be the lack of tracking for ovulation in each participant, as this is a primary marker of hormonal health in women of reproductive age. Due to the exploratory nature of the original study, we designed the methods to capture more general changes in both men and women, versus just women alone thus, given the exploratory nature of this inquiry and the unforeseen avenue it took, we acknowledge that a pre-study statistical power calculation was not conducted. Secondly, we used an instrument to capture menses that had not been previously validated or tested for reliability. More robust questionnaires (i.e., Ferriman-Gallway Index PCOS measure predictor) and wearable ovulation tracking (OvuCore by OvuSense) will be used in the future to capture precise physiological changes specific to reproductive health in women. The self-reported menses changes herein must be viewed with the understanding of participant self-perception and assessment relative to asynchronous participant menstrual cycles during bi-weekly study visits.

### Conclusion

Precisely-fed KD and LFDs induced clinically-significant weight loss, and created a more favorable body composition, BMI, lipid metabolism, and insulin-sensitivity after six-weeks. Nutritional ketosis, whether induced by the KD or augmented by exogenous ketones, influenced self-reported menses positively, independent of weight-loss and significantly compared to the LFD. The mechanism through which the KD modulated menses physiology in pre-menopausal women remains unclear. Continuing this line of work, particularly in women who may benefit the most from reversing insulin-resistance (i.e., PCOS patients), may unveil novel therapeutic roles of ketosis.

### Supporting information

**S1 File. Study protocol.**
(PDF)

**S2 File. CONSORT 2010 checklist of information to include when reporting a randomised trial\*.**
(DOC)

**S3 File.**
(DOCX)

**S4 File.**
(XLSX)

# Acknowledgments

The authors are grateful to all the participants who dedicated their time and effort to completing this study.

# Author Contributions

**Conceptualization:** Milene L. Brownlow, Annalouise O'Connor, Jeff S. Volek.

**Data curation:** Madison L. Kackley, Alex Buga, Teryn N. Sapper.

**Formal analysis:** Madison L. Kackley, Alex Buga, Christopher D. Crabtree, Bradley T. Robinson, Justen T. Stoner, Drew D. Decker.

**Funding acquisition:** Jeff S. Volek.

**Investigation:** Jeff S. Volek.

**Methodology:** Milene L. Brownlow, Annalouise O'Connor, Jeff S. Volek.

**Project administration:** Madison L. Kackley, Jeff S. Volek.

**Resources:** Jeff S. Volek.

**Supervision:** Jeff S. Volek.

**Validation:** Milene L. Brownlow, Annalouise O'Connor, Jeff S. Volek.

**Writing – original draft:** Madison L. Kackley, Alex Buga, Jeff S. Volek.

**Writing – review & editing:** Madison L. Kackley, Alex Buga, Milene L. Brownlow, Annalouise O'Connor, Loriana Soma, Jeff S. Volek.

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
