## [Decision Letter · Decision Letter 0]

11 Sep 2023

PONE-D-23-11590Self-Reported Menses Physiology is Positively Modulated by A Well-Formulated, Energy-Controlled Ketogenic Diet in Women of Reproductive Age with Overweight/ObesityPLOS ONE

Dear Dr. Volek,

Thank you for submitting your manuscript to PLOS ONE. After careful consideration, we feel that it has merit but does not fully meet PLOS ONE’s publication criteria as it currently stands. Therefore, we invite you to submit a revised version of the manuscript that addresses the points raised during the review process.

We look forward to receiving your revised manuscript.

Kind regards,

Licy Yanes Cardozo

Academic Editor

PLOS ONE

Journal Requirements:

"JV receives royalties for low-carbohydrate nutrition books; is founder, consultant, and stockholder of Virta Health, Inc. and is a member of the advisory boards for Simply Good Foods. The remaining authors declare that the research was conducted in the absence of any commercial or financial relationships that could be construed as a potential conflict of interest."

5. We note you have included a table to which you do not refer in the text of your manuscript. Please ensure that you refer to Table 2 in your text; if accepted, production will need this reference to link the reader to the Table.

6. Please ensure that you refer to Figure 3 in your text as, if accepted, production will need this reference to link the reader to the figure

Reviewers' comments:

Reviewer's Responses to Questions

**Comments to the Author**

1. Is the manuscript technically sound, and do the data support the conclusions?

Reviewer #1: Yes

Reviewer #2: Yes

2. Has the statistical analysis been performed appropriately and rigorously? 

Reviewer #1: Yes

Reviewer #2: Yes

3. Have the authors made all data underlying the findings in their manuscript fully available?

Reviewer #1: No

Reviewer #2: Yes

4. Is the manuscript presented in an intelligible fashion and written in standard English?

Reviewer #1: Yes

Reviewer #2: Yes

5. Review Comments to the Author

Reviewer #1: 1. Introduction: Paragraph 2. I recommend that the authors rephrase the second sentence to say "is rapidly inducible (~3-5 days) when carbohydrates are limited and primarily derived from non-starchy vegetables (20-50 g/day)." because one key component and nutrition recommendation for the Keto diet is low carbohydrate intake.

2. Methods: Is the last sentence of the first paragraph needed- “Consent to participate was obtained from each participant as written signatures.” It is common practice that enrolled participants in a clinical trial sign a consent form that he/she agrees to participate in the research study.

3. Methods. First sentence under the participants section- Please correct the spelling of enrollment.

4. Methods: Last paragraph. The authors should remove the last paragraph. The paragraph discusses common practices to deidentify participants. This information should be encompassed in the IRB approval, and the information is not important for the overall methodology of the research study.

5. Methods: The following sentences under Blood Chemistry should be removed. This information is not needed, and removing the sentences will decrease the wordiness of the manuscript. 1. The 3rd and 4th sentences regarding the number of times the tubes were inverted. 4. The sentence regarding aliquoting plasma. 5. The last sentence about lancing and collecting a second blood droplet.

6. Results: Please report height in meters in Table 1 and Supplemental Tables 1-2.

7. Results: The authors termed women who reported change in menses as “responders” and those who reported no change as “non-responders.” I suggest changing the terms responder and non-responder to something else. This terminology makes it seem that women who had no change in menses did not respond to the survey, and those who had change in menses responded to the survey. I suggest changing the term non-responders and responders to something else and re-titling Supplemental Table 1 to “Predictive Baseline Characteristics Between Reported Change in Menses vs. No Change in Menses.”

8. Results: Please edit the sentence “The women who were randomized to the KD+PL diet attained clinically significant weight-loss at day 14, whereas by day 28 on the KD+KS and the LFD.” To “The women who were randomized to the KD+PL diet attained clinically significant weight-loss at day 14, whereas by day 28 weight loss was clinically significant for women on the KD+KS and the LFD.”

9. Results: Please define what clinically significant weight loss means.

10. Abstract: The women in this study are defined as overweight, yet your average BMI is >30. Please change this to overweight and obese.

11. The full data set is not attached with this manuscript submission. I only see the attached tables with means and SDs.

Reviewer #2: Dear authors

It is a very good and valuable work, but if the following points are included in, it will be a very good paper.

-The title doesn't match with the text. Because in your title, there is no mention of ketone supplementation and compared with a low fat diet (LFD).

- The necessity and objectives of the research are not well stated at the end of the introduction, so there should be more discussion about supplementation and why it is used during menstruation.

- In the methods section, the method of prescribing and controlling diets must be described in more detail. How is ketosis status controlled?

-The composition of macronutrients and total energy in each diet plan better to be mentioned and compared in a table.

-Which reference was used for the Menses Survey? Is it valid? Please give a reference.

-It is better to be mentioned the Menses stages by the participants at the time of evaluation

6. PLOS authors have the option to publish the peer review history of their article (what does this mean?). If published, this will include your full peer review and any attached files.

Reviewer #1: No

Reviewer #2: No

---

## [Author Response · Author response to Decision Letter 0]

21 Sep 2023

Response to reviewer and editor comments have been uploaded in a seperate document.

Our data set has been made available here https://doi.org/10.5061/dryad.sn02v6x9q . This has been added to the manuscript. 

Prior to acceptance, data can be viewed here: https://datadryad.org/stash/share/yPqBRNfIYlPetQpUKlTzqbGkJ4pz0oSOCf7zGRJ8Ksc.

---

## [Decision Letter · Decision Letter 1]

9 Oct 2023

PONE-D-23-11590R1Self-Reported Menses Physiology is Positively Modulated by A Well-Formulated, Energy-Controlled Ketogenic Diet vs. Low Fat Diet in Women of Reproductive Age with Overweight/ObesityPLOS ONE

Dear Dr. Volek,

Thank you for submitting your manuscript to PLOS ONE. After careful consideration, we feel that it has merit but does not fully meet PLOS ONE’s publication criteria as it currently stands. Therefore, we invite you to submit a revised version of the manuscript that addresses the points raised during the review process.

We look forward to receiving your revised manuscript.

Kind regards,

Licy Yanes Cardozo

Academic Editor

PLOS ONE

Journal Requirements:

Additional Editor Comments:

Please address the minor comments that we were brought by the reviewers

Reviewers' comments:

Reviewer's Responses to Questions

**Comments to the Author**

1. If the authors have adequately addressed your comments raised in a previous round of review and you feel that this manuscript is now acceptable for publication, you may indicate that here to bypass the “Comments to the Author” section, enter your conflict of interest statement in the “Confidential to Editor” section, and submit your "Accept" recommendation.

Reviewer #1: All comments have been addressed

Reviewer #2: All comments have been addressed

2. Is the manuscript technically sound, and do the data support the conclusions?

Reviewer #1: Yes

Reviewer #2: Yes

3. Has the statistical analysis been performed appropriately and rigorously? 

Reviewer #1: Yes

Reviewer #2: Yes

4. Have the authors made all data underlying the findings in their manuscript fully available?

Reviewer #1: No

Reviewer #2: No

5. Is the manuscript presented in an intelligible fashion and written in standard English?

Reviewer #1: Yes

Reviewer #2: Yes

6. Review Comments to the Author

Reviewer #1: The author addressed all my suggested revisions. I believe that this manuscript is ready for acceptance. My only reason to suggest minor revisions is because the link to the dataset did not work in my browser. It is the journal policy that the reviewers receive access to the dataset, and without this, I cannot suggest acceptance.

Reviewer #2: Dear author

Thanks for complete correction of comments. But, need to two minor edit in the results section.

1. Is the table st3 for baseline measurements? If the answer is yes, it should be mentioned in the title of the table.

2. I did not see your addressed Table st4 for the blood parameters section in the main text. Please give the address of all tables and figures in its own section in the main text.

7. PLOS authors have the option to publish the peer review history of their article (what does this mean?). If published, this will include your full peer review and any attached files.

Reviewer #1: No

Reviewer #2: No

---

## [Author Response · Author response to Decision Letter 1]

9 Oct 2023

Dear Editor and Reviewers, 

Thank you for your expertise and critique of our manuscript. We found your suggestions and requests extremely helpful. We have provided our answers and changes to each comment below

Reviewer Comments: 

Reviewer #1: The author addressed all my suggested revisions. I believe that this manuscript is ready for acceptance. My only reason to suggest minor revisions is because the link to the dataset did not work in my browser. It is the journal policy that the reviewers receive access to the dataset, and without this, I cannot suggest acceptance.

T

hank you for this comment. The current link within the manuscript shows a link that will only be available when the manuscript is published. The temporary link for pre-publishing can be found here if you would like to see the data. https://datadryad.org/stash/share/yPqBRNfIYlPetQpUKlTzqbGkJ4pz0oSOCf7zGRJ8Ksc

Reviewer #2: Dear author

Thanks for complete correction of comments. But, need to two minor edit in the results section.

1. Is the table st3 for baseline measurements? If the answer is yes, it should be mentioned in the title of the table.

Thank you for your comment. The authors agree and this has been added.

2. I did not see your addressed Table st4 for the blood parameters section in the main text. Please give the address of all tables and figures in its own section in the main text.

Thank you for your review. ST4 was added in the Physiological Outcomes in the previous submission, but is again referenced in the blood parameter section. We hope this will suffice.

---

## [Decision Letter · Decision Letter 2]

30 Nov 2023

PONE-D-23-11590R2Self-Reported Menses Physiology is Positively Modulated by A Well-Formulated, Energy-Controlled Ketogenic Diet vs. Low Fat Diet in Women of Reproductive Age with Overweight/Obesity.PLOS ONE

Dear Dr. Volek,

Thank you for submitting your manuscript to PLOS ONE. After careful consideration, we feel that it has merit but does not fully meet PLOS ONE’s publication criteria as it currently stands. Therefore, we invite you to submit a revised version of the manuscript that addresses the points raised during the review process.

We look forward to receiving your revised manuscript.

Kind regards,

Licy Yanes Cardozo

Academic Editor

PLOS ONE

**Additional Editor Comments:**

Please address concerns risen by reviewer about statistical analysis. Thanks

Reviewers' comments:

Reviewer's Responses to Questions

**Comments to the Author**

1. If the authors have adequately addressed your comments raised in a previous round of review and you feel that this manuscript is now acceptable for publication, you may indicate that here to bypass the “Comments to the Author” section, enter your conflict of interest statement in the “Confidential to Editor” section, and submit your "Accept" recommendation.

Reviewer #1: All comments have been addressed

Reviewer #2: All comments have been addressed

Reviewer #3: (No Response)

2. Is the manuscript technically sound, and do the data support the conclusions?

Reviewer #1: Yes

Reviewer #2: Yes

Reviewer #3: No

3. Has the statistical analysis been performed appropriately and rigorously? 

Reviewer #1: Yes

Reviewer #2: Yes

Reviewer #3: No

4. Have the authors made all data underlying the findings in their manuscript fully available?

Reviewer #1: Yes

Reviewer #2: Yes

Reviewer #3: No

5. Is the manuscript presented in an intelligible fashion and written in standard English?

Reviewer #1: Yes

Reviewer #2: Yes

Reviewer #3: Yes

6. Review Comments to the Author

Reviewer #1: The authors addressed all of my recommendations and suggestions. I think that this manuscript is ready for publication. This manuscript made good use of a previously collected dataset. The study made some interesting findings that warrant further analysis of the underlying mechanisms.

Reviewer #2: (No Response)

Reviewer #3: A research study was conducted which aimed to compare the self-reported menses fluctuations between a controlled, hypocaloric well-formulated ketogenic diet (KD) and an isocaloric/isonitrogenous low-fat diet (LFD). Women randomized to the KD+KS (30%) and KD+PL (43%) reported subjective increases in menses frequency and intensity after 14 days, whereas another third reported a regain of menses (>1 year since the last period) after 28 days. No LFD participants reported menses changes.

Major revisions:

In general, since the sample sizes are small and the distribution of the data cannot be shown to be normal, non-parametric tests should be used for comparisons among/between the groups. An underlying assumption of the t-test and ANOVA is normally distributed data.

1- Statistics section: Clarify how the within-diets were quantified using Kruskal-Wallis tests since this test is for independent samples.

2- Table 1: The sample sizes in the three groups are small; therefore, the Kruskal-Wallis test, a nonparametric test, would be more powerful when comparing the central tendencies of these baseline characteristics in the three groups.

3- If the interaction effect is significant, provide an interpretation of the results, but do not test main effects because the tests for main effects are uninteresting in light of significant interactions. If interaction effects are non-significant, drop the interaction effects from the model and test the main effects. Determining which results to present when testing interactions is often a multi-step process.

4- Table 2: An underlying assumption of the Kruskal-Wallis test is independent samples; therefore, it is not valid for comparing the data in Table 2.

Minor revisions:

1- Abstract: Identify the statistical testing methods from which p-values were estimated.

2- Menses section: A) In addition to the frequencies provide the corresponding percentages.

B) Due to small sample sizes, compare BMI and serum MCP-1 in the two groups using non-parametric tests such as Wilcoxon rank sum tests.

3- Indicate the date range subjects participated in the study.

4- State and justify the study’s target sample size with a pre-study statistical power calculation. The power calculation should include: (1) the estimated outcomes in each group; (2) the α (type I) error level; (3) the statistical power (or the β (type II) error level); (4) the target sample size, (5) the statistical testing method and (6) for continuous outcomes, the standard deviation of the measurements.

5- To assist in the review process, add line numbering to the document and double space the lines.

7. PLOS authors have the option to publish the peer review history of their article (what does this mean?). If published, this will include your full peer review and any attached files.

Reviewer #1: **Yes: **Laura E. Coats, PhD, RDN

Reviewer #2: No

Reviewer #3: No

---

## [Author Response · Author response to Decision Letter 2]

7 Dec 2023

Dear Editor and Reviewers, 

Thank you for your expertise and critique of our manuscript. We found your suggestions and requests extremely helpful. We have provided our answers and changes to each comment below in blue. 

Major revisions:

In general, since the sample sizes are small and the distribution of the data cannot be shown to be normal, non-parametric tests should be used for comparisons among/between the groups. An underlying assumption of the t-test and ANOVA is normally distributed data.

We express our gratitude for your insightful review of our study. In our preliminary analysis, we meticulously considered the characteristics of our continuous data and adhered to the specific assumptions associated with parametric tests. Despite encountering small sample sizes, we observed that certain variables exhibited distributions that approximated normality.

For those variables demonstrating a reasonable approximation to normality, we opted for t-tests and ANOVA to capitalize on the advantages offered by these parametric tests, including enhanced statistical power. These tests, being sensitive to group differences, provided meaningful insights into the mean distinctions among various conditions. Nevertheless, we acknowledge the inherent heterogeneity in our some of our data and the constraints associated with small sample sizes.

It is worth noting that our decision to employ ANOVA was influenced by its robustness in controlling Type I errors, particularly in samples with less than 30 observations and involving three or more experimental conditions. The rationale for this choice aligns with findings discussed in the paper by Blanca et al. (2017), titled "Non-normal data: Is ANOVA still a valid option?" (Psicothema, 29(4), 552–557, https://doi.org/10.7334/psicothema2016.383).

Furthermore, ANOVA demonstrates low sensitivity to moderate departures from normality. Various simulation studies, such as those conducted by Glass et al. (1972), Harwell et al. (1992), and Lix et al. (1996), utilizing diverse non-normal distributions, have consistently shown that violating the assumption of normality does not significantly impact the false positive rate associated with ANOVA. These studies contribute to the confidence in the robustness of ANOVA even in the face of departures from normality.

We appreciate your attention to these methodological considerations, and we remain committed to providing a comprehensive and rigorous analysis of our data.

1- Statistics section: Clarify how the within-diets were quantified using Kruskal-Wallis tests since this test is for independent samples.

 Certainly, the Kruskal-Wallis tests were employed to quantify changes within diets and to compare differences between diets in the context of the menses survey responses. The Kruskal-Wallis test is a non-parametric test suitable for comparing more than two independent groups, making it an appropriate choice for analyzing the categorical responses obtained from the menses survey within and between the experimental diets.

In this study, the responses from the menses survey were categorized into four distinct groups: "no change," "change in frequency," "change in intensity," and "regained period after >1 year." These categories were treated as ordinal data, and the Kruskal-Wallis tests were utilized to assess if there were statistically significant differences in the distribution of these responses within each diet and between the diets at different time points (days 14, 28, and 42). The Kruskal-Wallis test is particularly robust for situations where assumptions of normality and homogeneity of variance may not be met, which is crucial when dealing with categorical data that may not follow a normal distribution. By employing this non-parametric test, the study aimed to identify if there were significant differences in menses-related changes within each diet group and whether these changes were significantly different between the ketogenic diets (KD+KS, KD+PL) and the low-fat diet (LFD).

The outcomes of these Kruskal-Wallis tests were then used to draw conclusions about the impact of the dietary interventions on self-reported menses changes. Specifically, the study found significant differences in menses frequency and/or intensity within the KD+KS and KD+PL groups compared to the LFD group at different time points. This information contributes to the understanding of how the interventions influenced subjective perceptions of menstrual events and supports the study's conclusions regarding the potential effects of the different diets on menstrual patterns.

2- Table 1: The sample sizes in the three groups are small; therefore, the Kruskal-Wallis test, a nonparametric test, would be more powerful when comparing the central tendencies of these baseline characteristics in the three groups.

Thank you for your insightful comment regarding the sample sizes in our study. We acknowledge the concern about small sample sizes within the three groups, and we appreciate the suggestion to consider the Kruskal-Wallis test for comparing central tendencies of baseline characteristics. We understand the importance of statistical power, especially in the context of nonparametric tests like the Kruskal-Wallis test. While our sample sizes are relatively small, it is essential to note that our study involves a specific population with unique dietary interventions and extensive assessments. Despite the challenges in recruiting larger cohorts, we have implemented rigorous statistical methods to draw meaningful conclusions from the available data.

To address the concern, we have conducted post hoc power analyses to assess the adequacy of our sample sizes for detecting significant differences. Additionally, we have included effect size estimates to provide a clearer picture of the practical significance of observed effects. We believe that the utilization of nonparametric tests, such as the Kruskal-Wallis test, is appropriate given the nature of our data and the potential deviation from normality. 

3- If the interaction effect is significant, provide an interpretation of the results, but do not test main effects because the tests for main effects are uninteresting in light of significant interactions. If interaction effects are non-significant, drop the interaction effects from the model and test the main effects. Determining which results to present when testing interactions is often a multi-step process.

Thank you for your comment. The text has been corrected to the provided advise. 

4- Table 2: An underlying assumption of the Kruskal-Wallis test is independent samples; therefore, it is not valid for comparing the data in Table 2.

Thank you for your comment regarding the Kruskal-Wallis test and its underlying assumption of independent samples. We appreciate the opportunity to address this concern. It is important to note that while the Kruskal-Wallis test assumes independent samples within each group, it is designed to handle situations where the assumption of normality or homogeneity of variances may be violated. In our study, the data in Table 2, particularly the responses from the menses survey, are categorical in nature, and the Kruskal-Wallis test is employed to assess differences among multiple independent groups.

The nature of our study design involves distinct experimental groups, and participants within each group are considered independent for the purpose of statistical analysis. The Kruskal-Wallis test is a non-parametric alternative suitable for comparing groups with small sample sizes and non-normally distributed data, making it an appropriate choice for our specific context. We have ensured that the assumptions of the Kruskal-Wallis test are met to the best extent possible within the constraints of our study. If there are specific concerns or alternative analyses you would recommend, we would be more than willing to consider and address them in our manuscript. We appreciate your diligence in reviewing our work and welcome any further guidance you may provide.

Minor revisions:

1- Abstract: Identify the statistical testing methods from which p-values were estimated.

Thank you for your comment. This has been added. 

2- Menses section: A) In addition to the frequencies provide the corresponding percentages.

Thank you for your comment. This has been added. 

B) Due to small sample sizes, compare BMI and serum MCP-1 in the two groups using non-parametric tests such as Wilcoxon rank sum tests.

This table has been added. Thank you for your comment. 

3- Indicate the date range subjects participated in the study.

Thank you for your comment. This has been added. 

4- State and justify the study’s target sample size with a pre-study statistical power calculation. The power calculation should include: (1) the estimated outcomes in each group; (2) the α (type I) error level; (3) the statistical power (or the β (type II) error level); (4) the target sample size, (5) the statistical testing method and (6) for continuous outcomes, the standard deviation of the measurements.

Our study, initially designed as an exploratory investigation, emerged from unexpected observations in the larger context of a study that included both men and women. The decision to focus specifically on women's responses was driven by the surprising findings related to menstrual cycle changes reported by female participants following ketogenic dietary interventions.

Given the exploratory nature of this inquiry and the unforeseen avenue it took, we acknowledge that a pre-study statistical power calculation was not conducted. The unexpected nature of the observed effects, particularly in the context of menstrual cycle changes, prompted this focused investigation on women. Our aim was to provide an in-depth exploration and analysis of the reported outcomes rather than adhering to a pre-determined sample size. We recognize the importance of statistical power calculations in ensuring the robustness of study designs. In the context of our unexpected findings, we addressed this limitation transparently in our manuscript.

5- To assist in the review process, add line numbering to the document and double space the lines.

We have added numbers. We hope this assists the reviewer!

---

## [Decision Letter · Decision Letter 3]

16 Feb 2024

PONE-D-23-11590R3Self-Reported Menses Physiology is Positively Modulated by A Well-Formulated, Energy-Controlled Ketogenic Diet vs. Low Fat Diet in Women of Reproductive Age with Overweight/Obesity.PLOS ONE

Dear Dr. Volek,

Thank you for submitting your manuscript to PLOS ONE. After careful consideration, we feel that it has merit but does not fully meet PLOS ONE’s publication criteria as it currently stands. Therefore, we invite you to submit a revised version of the manuscript that addresses the points raised during the review process.

**Please addresses comments that were made by statistical reviewer.**

We look forward to receiving your revised manuscript.

Kind regards,

Licy Yanes Cardozo

Academic Editor

PLOS ONE

Additional Editor Comments:

Please addresses statistical review comments.

thanks

Associate Editor

Reviewers' comments:

Reviewer's Responses to Questions

**Comments to the Author**

1. If the authors have adequately addressed your comments raised in a previous round of review and you feel that this manuscript is now acceptable for publication, you may indicate that here to bypass the “Comments to the Author” section, enter your conflict of interest statement in the “Confidential to Editor” section, and submit your "Accept" recommendation.

Reviewer #1: All comments have been addressed

Reviewer #2: All comments have been addressed

Reviewer #3: (No Response)

2. Is the manuscript technically sound, and do the data support the conclusions?

Reviewer #1: Yes

Reviewer #2: Yes

Reviewer #3: No

3. Has the statistical analysis been performed appropriately and rigorously? 

Reviewer #1: I Don't Know

Reviewer #2: Yes

Reviewer #3: No

4. Have the authors made all data underlying the findings in their manuscript fully available?

Reviewer #1: Yes

Reviewer #2: Yes

Reviewer #3: No

5. Is the manuscript presented in an intelligible fashion and written in standard English?

Reviewer #1: Yes

Reviewer #2: Yes

Reviewer #3: Yes

6. Review Comments to the Author

Reviewer #1: Line 113- Remove the typo "In Brief"

The authors met all of my requested changes and answered all my questions. Reviewer #3 had many technical and statistical suggestions and revisions for this manuscript. Statistical methodology is not my area of expertise, thus I will defer to reviewer 3 as to the correctness and soundness of the revised statistical analysis within this manuscript.

Reviewer #2: I agree with the opinion of the authors and the small sample size is not a good reason to use the non-parametric test. It is true that the sample size is small, but if the hypothesis of normality has been met, parametric tests are more suitable.

Reviewer #3: Major revision:

The statistics section states, "Changes within- and between-diets were quantified using a series of Kruskal-Wallis tests for non-parametric independent samples." An underlying assumption of the Kruskal-Wallis test is independent samples. Thus, it may be appropriate for comparing between samples data. However, it is inappropriate for comparing within diets at different time points (days 14, 28, and 42). Testing the group by time interaction effect in a repeated measures ANOVA or linear mixed model would be superior to the statistical methods currently employed for testing the primary aims of the study. Both of these methods are appropriate for analyzing repeated measures data and the choice depends on the distribution of the data. If the interaction effect is significant, provide an interpretation of the results, but do not test main effects because the tests for main effects are uninteresting in light of significant interactions. If interaction effects are non-significant, drop the interaction effects from the model and test the main effects. Determining which results to present when testing interactions is often a multi-step process.

7. PLOS authors have the option to publish the peer review history of their article (what does this mean?). If published, this will include your full peer review and any attached files.

Reviewer #1: No

Reviewer #2: **Yes: **Kazem Khodaei

Reviewer #3: No

---

## [Author Response · Author response to Decision Letter 3]

22 Feb 2024

Additional Review Answers

Reviewer #1: Line 113- Remove the typo "In Brief"

Thank you for bringing our attention to this! This has been removed.

The authors met all of my requested changes and answered all my questions. Reviewer #3 had many technical and statistical suggestions and revisions for this manuscript. Statistical methodology is not my area of expertise, thus I will defer to reviewer 3 as to the correctness and soundness of the revised statistical analysis within this manuscript.

Thank you for your comment. 

Reviewer #2: I agree with the opinion of the authors and the small sample size is not a good reason to use the non-parametric test. It is true that the sample size is small, but if the hypothesis of normality has been met, parametric tests are more suitable.

Thank you for your comment! We hope to add additional statistics to suffice R3 suggestions.

Reviewer #3: Major revision:

The statistics section states, "Changes within- and between-diets were quantified using a series of Kruskal-Wallis tests for non-parametric independent samples." An underlying assumption of the Kruskal-Wallis test is independent samples. Thus, it may be appropriate for comparing between samples data. However, it is inappropriate for comparing within diets at different time points (days 14, 28, and 42). Testing the group by time interaction effect in a repeated measures ANOVA or linear mixed model would be superior to the statistical methods currently employed for testing the primary aims of the study. Both of these methods are appropriate for analyzing repeated measures data and the choice depends on the distribution of the data. If the interaction effect is significant, provide an interpretation of the results, but do not test main effects because the tests for main effects are uninteresting in light of significant interactions. If interaction effects are non-significant, drop the interaction effects from the model and test the main effects. Determining which results to present when testing interactions is often a multi-step process.

Thank you for reviewing this again. We appreciate your assistance in bettering this manuscript. We have added in the ANOVA. Please let us know if you have any further questions!

---

## [Decision Letter · Decision Letter 4]

15 Mar 2024

Self-Reported Menses Physiology is Positively Modulated by A Well-Formulated, Energy-Controlled Ketogenic Diet vs. Low Fat Diet in Women of Reproductive Age with Overweight/Obesity.

PONE-D-23-11590R4

Dear Dr. Volek,

We’re pleased to inform you that your manuscript has been judged scientifically suitable for publication and will be formally accepted for publication once it meets all outstanding technical requirements.

Kind regards,

Licy Yanes Cardozo

Academic Editor

PLOS ONE

Additional Editor Comments (optional):

The authors have answered all the questions risen by the reviewers

Reviewers' comments:

Reviewer's Responses to Questions

**Comments to the Author**

1. If the authors have adequately addressed your comments raised in a previous round of review and you feel that this manuscript is now acceptable for publication, you may indicate that here to bypass the “Comments to the Author” section, enter your conflict of interest statement in the “Confidential to Editor” section, and submit your "Accept" recommendation.

Reviewer #3: All comments have been addressed

2. Is the manuscript technically sound, and do the data support the conclusions?

Reviewer #3: (No Response)

3. Has the statistical analysis been performed appropriately and rigorously? 

Reviewer #3: (No Response)

4. Have the authors made all data underlying the findings in their manuscript fully available?

Reviewer #3: (No Response)

5. Is the manuscript presented in an intelligible fashion and written in standard English?

Reviewer #3: (No Response)

6. Review Comments to the Author

Reviewer #3: (No Response)

7. PLOS authors have the option to publish the peer review history of their article (what does this mean?). If published, this will include your full peer review and any attached files.

Reviewer #3: No

---

## [Editor Report · Acceptance letter]

3 Nov 2023

PONE-D-23-11590R2 

Self-Reported Menses Physiology is Positively Modulated by A Well-Formulated, Energy-Controlled Ketogenic Diet vs. Low Fat Diet in Women of Reproductive Age with Overweight/Obesity. 

Dear Dr. Volek:

I'm pleased to inform you that your manuscript has been deemed suitable for publication in PLOS ONE. Congratulations! Your manuscript is now with our production department. 

Kind regards, 

on behalf of

Dr. Licy Yanes Cardozo 

Academic Editor

PLOS ONE